# Learning SMaLL Predictors

**Vikas K. Garg**
CSAIL, MIT
vgarg@csail.mit.edu

**Ofer Dekel**
Microsoft Research
oferd@microsoft.com

**Lin Xiao**
Microsoft Research
lin.xiao@microsoft.com

## Abstract

We introduce a new framework for learning in severely resource-constrained settings. Our technique delicately amalgamates the representational richness of multiple linear predictors with the sparsity of Boolean relaxations, and thereby yields classifiers that are compact, interpretable, and accurate. We provide a rigorous formalism of the learning problem, and establish fast convergence of the ensuing algorithm via relaxation to a minimax saddle point objective. We supplement the theoretical foundations of our work with an extensive empirical evaluation.

## 1  Introduction

Modern advances in machine learning have produced models that achieve unprecedented accuracy on standard prediction tasks. However, this remarkable progress in model accuracy has come at a significant cost. Many state-of-the-art models have ballooned in size and applying them to a new point can require tens of GFLOPs, which renders these methods ineffectual on resource-constrained platforms like smart phones and wearables [1, 2]. Indeed, in these settings, inference with a compact learner that can fit on the small device becomes an overarching determinant even if it comes at the expense of slightly worse accuracy. Moreover, large models are often difficult to interpret, simply because humans are not good at reasoning about large, complex objects. Modern machine learning models are also more costly to train, but we sidestep that problem in this paper by assuming that we can train our models on powerful servers in the cloud.

In our pursuit of compact and interpretable models, we take inspiration from the classic problem of learning *disjunctive normal forms* (DNFs) [3]. Specifically, a $p$-term $k$-DNF is a DNF with $p$ terms, where each term contains exactly $k$ Boolean variables. Small DNFs are a natural starting point for our research, because they pack a powerful nonlinear descriptive capacity in a succinct form. The DNF structure is also known to be intuitive and interpretable by humans [4, 5]. However, with the exception of a few practical heuristics [4, 5, 6], an overwhelming body of work [7, 8, 9, 10, 11, 12, 13, 14, 15, 16, 17] theoretically characterizes the difficulty of learning a $k$-DNF under various restricted models of learning. Our method, *Sparse Multiprototype Linear Learner* (SMaLL), bypasses this issue by crafting a continuous relaxation that amounts to a form of *improper* learning of the $k$-DNFs in the sense that the hypothesis space subsumes $p$-term $k$-DNF classifiers, and thus is at least as powerful as the original $k$-DNF family. Armed with our technical paraphernalia, we design a practical algorithm that yields small and interpretable models.

Our work may also be viewed as a delicate fusion of multiple prototypes [1, 18, 19, 20, 21, 22] with Boolean relaxations [23]. The richness of models with multiple prototypes overcomes the representational limitations of sparse linear models like Lasso and Elastic-Net [24, 25, 26] that are typically not expressive enough to achieve state-of-the-art accuracy. Boolean relaxations afford us the ability to control the degree of sparsity explicitly in our predictors akin to exploiting an $\ell_0$ regularization, unlike the $\ell_1$ based methods that may require extensive tuning. Thus, our approach harnesses the best of both worlds. Moreover, folding sparsity in the training objective obviates the costs that would otherwise be incurred in compressing a large model via methods like pruning [27, 28, 29], low-rank approximation [30, 31], hashing [32], or parameter quantization [27, 33].

Additionally, we overcome some significant limitations of other methods that use a small number of prototypes, such as [1, 19, 34]. These techniques invariably require solving highly non-convex or combinatorially hard mixed integer optimization problems, which makes it difficult to guarantee their convergence and optimality. We derive a minimax saddle-point relaxation that provably admits $O(1/t)$ convergence via our customized Mirror-Prox algorithm. We provide detailed empirical results that demonstrate the benefits of our approach on a large variety of OpenML datasets. Specifically, on many of these datasets, our algorithm either surpasses the accuracy of the state-of-the-art baselines, or provides more compact models while being competent in terms of accuracy.

In Section 2, we formulate the problem of learning a $k$-sparse $p$-prototype linear predictor as a mixed integer nonlinear optimization problem. Then, in Section 3, we relax this optimization problem to a saddle-point problem, which we solve using a Mirror-Prox algorithm. Finally, we present empirical results in Section 4. All the proofs are provided in the Supplementary to keep the exposition focused.

## 2   Problem Formulation

We first derive a convex loss function for multiprototype binary classification. Let $\{(x_i, y_i)\}_{i=1}^m$ be a training set of instance-label pairs, where each $x_i \in \mathbb{R}^n$ and each $y_i \in \{-1, 1\}$. Let $\ell : \mathbb{R} \mapsto \mathbb{R}$ be a convex surrogate for the error indicator function $\mathbb{1}_{f(x_i) \neq y_i} = 1$ if $y_i f(x_i) < 0$ and 0 otherwise. We also assume that $\ell$ upper bounds the error indicator function and is monotonically non-increasing. In particular, the popular hinge-loss and log-loss functions satisfy these properties.

Let $\{w_j\}_{j=1}^p$ be a set of linear prototypes. We consider a binary classifier of the form

$$f(x) = \text{sign}\left(\max_{j \in [p]} w_j \cdot x\right) .$$

Our decision rule is motivated by the following result.

**Proposition 1.** *Consider the class $\mathcal{C}_k = \{(w_1, w_2, \ldots, w_p) | \forall j \in [p], w_j \in \mathbb{R}^n, ||w_j||_0 = k\}$ of $p$ prototypes, where each prototype is $k$-sparse for $k \geq 0$. For any $x \in \mathbb{R}^n$, let the predictors $f = (w_1, w_2, \ldots, w_p) \in \mathcal{C}_k$ take the following form:*

$$f(x) = 1 \text{ if } \max_{j \in [p]} w_j \cdot x \geq k, \text{ and } -1 \text{ otherwise.}$$

*Learning $\mathcal{C}_k$ amounts to improper learning of $p$-term $k$-DNF Boolean formulae.*

Thus, our search space contains the family of $k$-DNF classifiers, though owing to the hardness of learning $k$-DNF, we may not always find a $k$-DNF classifier. Nonetheless, due to improper learning, the value of the objective returned will be a lower bound on the cost objective achieved by the space of $p$-term $k$-DNF classifiers (much like the relation between an integer program and its relaxation).

We handle the negative and positive examples separately. For each negative training example $(x_i, -1)$, the classifier makes a correct prediction if and only if $\max_{j \in [p]} w_j \cdot x_i < 0$. Under our assumptions on $\ell$, the error indicator function can be upper bounded as

$$\mathbb{1}_{f(x_i) \neq -1} \leq \ell\left(-\max_{j \in [p]} w_j \cdot x_i\right) = \max_{j \in [p]} \ell(-w_j \cdot x_i),$$

where the equality holds because we assume that $\ell$ is monotonically non-increasing. We note that the upper bound $\max_{j \in [p]} \ell(-w_j \cdot x_i)$ is jointly convex in $\{w_j\}_{j=1}^p$ [35, Section 3.2.3].

For each positive example $(x_i, +1)$, the classifier makes a correct prediction if and only if $\max_{j \in [p]} w_j \cdot x_i > 0$. By our assumptions on $\ell$, we have

$$\mathbb{1}_{f(x_i) \neq +1} \leq \ell\left(\max_{j \in [p]} w_j \cdot x_i\right) = \min_{j \in [p]} \ell(w_j \cdot x_i). \tag{1}$$

Again, the equality above is due to the monotonic non-increasing property of $\ell$. Here the right-hand side $\min_{j \in [p]} \ell(w_j \cdot x_i)$ is not convex in $\{w_j\}_{j=1}^p$. We resolve this by designating a dedicated prototype $w_{j(i)}$ for each positive training example $(x_i, +1)$, and using the upper bound

$$\mathbb{1}_{f(x_i) \neq +1} \leq \ell(w_{j(i)} \cdot x_i).$$

In the extreme case, we can associate each positive example with a distinct prototype. Then there will be no loss of using $\ell(w_{j(i)} \cdot x_i)$ compared with the upper bound in (1) when we set $j(i) = \arg\max_{j \in [p]} w_j \cdot x_i$. However, in this case, the number of prototypes $p$ is equal to the number of positive examples, which can be excessively large for storage and computation as well as cause overfitting. In practice, we may cluster the positive examples into $p$ groups, where $p$ is much smaller than the number of positive examples, and assign all positive examples in each group with a common prototype. In other words, we have $j(i) = j(k)$ if $x_i$ and $x_k$ belong to the same cluster. This clustering step helps us provide a fast parametric alternative to the essentially non-parametric setting that assumes one prototype per positive example.

Overall, we have the following convex surrogate for the total number of training errors:

$$h(w_1, \ldots, w_p) = \sum_{i \in I_+} \ell\big(w_{j(i)} \cdot x_i\big) + \sum_{i \in I_-} \max_{j \in [p]} \ell(-w_j \cdot x_i), \tag{2}$$

where $I_+ = \{i : y_i = +1\}$ and $I_- = \{i : y_i = -1\}$. In the rest of this paper, we let $W \in \mathbb{R}^{p \times n}$ be the matrix formed by stacking the vectors $w_1^T, \ldots, w_p^T$ vertically, and denote the above loss function by $h(W)$. In order to train a multi-prototype classifier, we minimize the regularized surrogate loss:

$$\min_{W \in \mathbb{R}^{p \times n}} \frac{1}{m} h(W) + \frac{\lambda}{2} \|W\|_F^2, \tag{3}$$

where $\| \cdot \|_F$ denotes the Frobenius norm of a matrix.

## 2.1 Smoothing the Loss via Soft-Max

In this paper, we focus on the log-loss $\ell(z) = \log(1 + \exp(-z))$. Although this $\ell$ is a smooth function, the overall loss $h$ defined in (2) is non-smooth, due to the max operator in the sum over the set $I_-$. In order to take advantage of fast algorithms for smooth convex optimization, we smooth the loss function using soft-max. More specifically, we replace the non-smooth terms $\max_{j \in [p]} \ell(t_j)$ in (2) with the soft-max operator over $p$ items:

$$u(t) \triangleq \log\left(1 + \sum_{j \in [p]} \exp(-t_j)\right), \tag{4}$$

where $t = (t_1, \ldots, t_p) \in \mathbb{R}^p$. Then we obtain the smoothed loss function

$$\tilde{h}(W) = \sum_{i \in I_+} \ell\big(w_{j(i)} \cdot x_i\big) + \sum_{i \in I_-} u(W x_i), \tag{5}$$

around which we will customize our algorithm design. Next, we incorporate sparsity constraints explicitly for the prototypes $w_1, \ldots, w_p$.

## 2.2 Incorporating Sparsity via Binary Variables

With some abuse of notation, we let $\|w_j\|_0$ denote the number of non-zero entries of the vector $w_j$, and define

$$\|W\|_{0,\infty} \triangleq \max_{j \in [p]} \|w_j\|_0.$$

The requirement that each prototype be $k$-sparse translates into the constraint $\|W\|_{0,\infty} \le k$. Therefore the problem of training a SMaLL model with budget $k$ (for each prototype) can be formulated as

$$\min_{\substack{W \in \mathbb{R}^{p \times n} \\ \|W\|_{0,\infty} \le k}} \frac{1}{m} \tilde{h}(W) + \frac{\lambda}{2} \|W\|_F^2, \tag{6}$$

where $\tilde{h}$ is defined in (5). This is a very hard optimization problem due to the nonconvex sparsity constraint. In order to derive a convex relaxation, we follow the approach of [23] (cf. [36]) to introduce a binary matrix $\epsilon \in \{0,1\}^{p \times n}$ and rewrite (6) as

$$\min_{\substack{W \in \mathbb{R}^{p \times n} \\ \epsilon \in \{0,1\}^{p \times n}, \|\epsilon\|_{1,\infty} \le k}} \frac{1}{m} \tilde{h}(W \odot \epsilon) + \frac{\lambda}{2} \|W \odot \epsilon\|_F^2,$$

where $\odot$ denotes the Hadamard (i.e. entry-wise) product of two matrices. Here we have

$$\|\epsilon\|_{1,\infty} = \max_{j\in[p]} \|\epsilon_j\|_1 ,$$

where $\epsilon_j$ is the $j$th row of $\epsilon$. Since all entries of $\epsilon$ belong to $\{0, 1\}$, the constraint $\|\epsilon\|_{1,\infty} \leq k$ is the same as $\|\epsilon\|_{0,\infty} \leq k$. Noting that we can take $W_{ij} = 0$ when $\epsilon_{ij} = 0$ and vice-versa, this problem is equivalent to

$$\min_{\substack{W\in\mathbb{R}^{p\times n} \\ \epsilon\in\{0,1\}^{p\times n}, \|\epsilon\|_{1,\infty}\leq k}} \frac{1}{m}\tilde{h}(W \odot \epsilon) + \frac{\lambda}{2}\|W\|_F^2 . \tag{7}$$

Using (5), the objective function can be written as

$$\frac{1}{m}\left( \sum_{i\in I_+} \ell\big((W \odot \epsilon)_{j(i)}x_i\big) + \sum_{i\in I_-} u\big(-(W \odot \epsilon)x_i\big)\right) + \frac{\lambda}{2}\|W\|_F^2,$$

where $(W \odot \epsilon)_{j(i)}$ denotes the $j(i)$th row of $W \odot \epsilon$.

So far our transformations have not changed the nature of the optimization problem with sparsity constraints — it is still a hard mixed-integer nonlinear optimization problem. However, as we will show in the next section, the introduction of the binary matrix $\epsilon$ allows us to derive a saddle-point formulation of problem (7), which in turn admits a convex-concave relaxation that can be solved efficiently by the Mirror-Prox algorithm [37, 38].

# 3 Saddle-Point Relaxation

We first show that the problem in (7) is equivalent to the following minimax saddle-point problem:

$$\min_{\substack{W\in\mathbb{R}^{p\times n} \\ \epsilon\in\{0,1\}^{p\times n}, \|\epsilon\|_{1,\infty}\leq k}} \max_{\substack{S=[s_1\cdots s_m] \\ s_i\in\mathcal{S}_i, i\in[m]}} \Phi(W, \epsilon, S), \tag{8}$$

where $S \in \mathbb{R}^{p\times m}$, each of its column $s_i$ belongs to a set $\mathcal{S}_i \subset \mathbb{R}^p$ (which will be given in Proposition 2), and the function $\Phi$ is defined as

$$\Phi(W, \epsilon, S) = \frac{1}{m}\sum_{i\in[m]}\Big(y_i s_i^T (W \odot \epsilon)x_i - u^\star(s_i)\Big) + \frac{\lambda}{2}\|W\|_F^2.$$

In the above definition, $u^\star$ is the convex conjugate of $u$ defined in (4):

$$u^\star(s_i) = \sup_{t\in\mathbb{R}^p} \big\{s_i^T t - u(t)\big\} \tag{9}$$

$$= \begin{cases} \sum_{j=1}^p (-s_{i,j}) \log(-s_{i,j}) + (1+\mathbf{1}^T s_i) \log(1+\mathbf{1}^T s_i), & \text{if } s_{i,j} \leq 0 \ \forall j \in [p] \text{ and } \mathbf{1}^T s_i \geq -1, \\ \infty, & \text{otherwise .} \end{cases}$$

The equivalence between (7) and (8) is a direct consequence of the following proposition.

**Proposition 2.** *Let $\ell(z) = \log(1 + \exp(-z))$ where $z \in \mathbb{R}$ and $u(t) = \log\big(1 + \sum_{j\in[p]} \exp(-t_j)\big)$ where $t \in \mathbb{R}^p$. Then for $i \in I_+$, we have*

$$\ell\big((W \odot \epsilon)_{j(i)}x_i\big) = \max_{s_i\in\mathcal{S}_i} \big(y_i s_i^T (W \odot \epsilon)x_i - u^\star(s_i)\big),$$

*where $y_i = +1$ and $\mathcal{S}_i = \big\{s_i \in \mathbb{R}^p : s_{i,j(i)} \in [-1,0], \ s_{i,j}=0 \ \forall j \neq j(i)\big\}$. For $i \in I_-$, we have*

$$u\big(-(W \odot \epsilon)x_i\big) = \max_{s_i\in\mathcal{S}_i} \big(y_i s_i^T (W \odot \epsilon)x_i - u^\star(s_i)\big),$$

*where $y_i = -1$ and $\mathcal{S}_i = \big\{s_i \in \mathbb{R}^p : \mathbf{1}^T s \geq -1, \ s_{i,j} \leq 0 \ \forall j \in [p]\big\}$.*

We can further eliminate the variable $W$ in (8). This is facilitated by the following result.

**Proposition 3.** *For any given $\epsilon \in \{0, 1\}^{p\times n}$ and $S \in \mathcal{S}_1 \times \cdots \times \mathcal{S}_m$, the solution to*

$$\min_{W\in\mathbb{R}^{p\times n}} \Phi(W, \epsilon, S)$$

*is unique and given by*

$$W(\epsilon, S) = -\frac{1}{m\lambda} \sum_{i\in[m]} y_i\big(s_i x_i^T\big) \odot \epsilon. \tag{10}$$

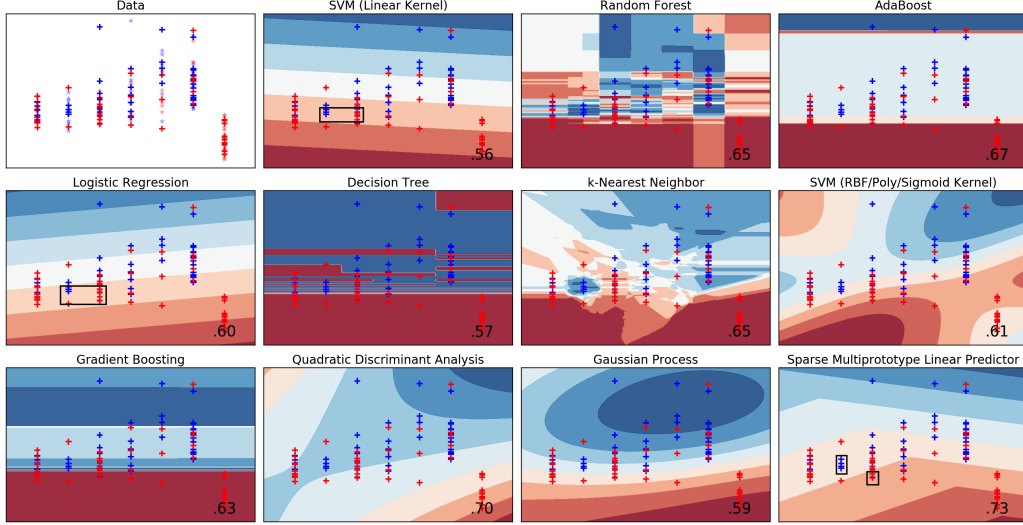

Figure 1: Decision surfaces of different classifier types on a run of the two-dimensional *chscase funds* toy dataset. Test classification accuracy is shown at the bottom right of each plot.

Now we substitute $W(\epsilon, S)$ into (8) to obtain

$$\min_{\substack{\epsilon \in \{0,1\}^{p \times n} \\ \|\epsilon\|_{1,\infty} \le k}} \max_{\substack{S = [s_1 \cdots s_m] \\ s_i \in \mathcal{S}_i,\ i \in [m]}} \phi(\epsilon, S)\,, \tag{11}$$

where

$$\phi(\epsilon, S) = -\frac{1}{m\lambda} \left\| \sum_{i \in [m]} y_i \left( s_i x_i^T \right) \odot \epsilon \right\|_F^2 - \sum_{i \in [m]} u^\star(s_i)\,.$$

We note that $\phi(\epsilon, S)$ is concave in $S$ (which is to be maximized), but not convex in $\epsilon$ (which is to be minimized). However, because $\epsilon \in \{0,1\}^{p \times n}$, we have $\epsilon \odot \epsilon = \epsilon$ and thus

$$\left\| \sum_{i \in [m]} y_i \left( s_i x_i^T \right) \odot \epsilon \right\|_F^2 = \sum_{i \in [m]} y_i s_i^T \left( \sum_{i \in [m]} y_i \left( s_i x_i^T \right) \odot \epsilon \odot \epsilon \right) x_i = \sum_{i \in [m]} y_i s_i^T \left( \sum_{i \in [m]} y_i \left( s_i x_i^T \right) \odot \epsilon \right) x_i\,.$$

Therefore the objective function $\phi$ in (11) can be written as

$$\phi(\epsilon, S) = -\frac{1}{m\lambda} \sum_{i \in [m]} y_i s_i^T \left( \sum_{i \in [m]} y_i \left( s_i x_i^T \right) \odot \epsilon \right) x_i - \sum_{i \in [m]} u^\star(s_i)\,, \tag{12}$$

which is concave in $S$ and linear (thus convex) in $\epsilon$.

Finally, we relax the integrality constraint on $\epsilon$ to its convex hull, i.e., $\epsilon \in [0,1]^{p \times n}$, and consider

$$\min_{\substack{\epsilon \in [0,1]^{p \times n} \\ \|\epsilon\|_{1,\infty} \le k}} \max_{\substack{S = [s_1 \cdots s_m] \\ s_i \in \mathcal{S}_i,\ i \in [m]}} \phi(\epsilon, S)\,, \tag{13}$$

where $\phi(\epsilon, S)$ is given in (12). This is a convex-concave saddle-point problem, which can be solved efficiently, for example, by the Mirror-Prox algorithm [37, 38].

After finding a solution $(\epsilon, S)$ of the relaxed problem (13), we can round the entries of $\epsilon$ to $\{0,1\}$, while respecting the constraint $\|\epsilon\|_{1,\infty} \le k$ (e.g., by rounding the largest $k$ entries of each row to 1 and the rest entries to 0, or randomized rounding). Then we can recover the prototypes using (10).

### 3.1  The Mirror-Prox Algorithm

Algorithm 1 lists the Mirror-Prox algorithm customized for solving the convex-concave saddle-point problem (13), which enjoys a $O(1/t)$ convergence rate [37, 38].

Table 1: Comparison of test accuracy on low dimensional ($n < 20$) OpenML datasets. $K$, in SMaLL, was set to $n$ for these datasets.

| | LSVM | RF | AB | LR | DT | kNN | RSVM | GB | GP | SMaLL |
|---|---|---|---|---|---|---|---|---|---|---|
| **bankruptcy** | .84±.07 | .83±.08 | .82±.05 | .90±.05 | .80±.05 | .78±.07 | .89±.06 | .81±.05 | .90±.05 | **.92±.06** |
| **vineyard** | .79±.10 | .72±.06 | .68±.04 | .82±.08 | .69±.13 | .70±.11 | .82±.07 | .68±.09 | .71±.12 | **.83±.07** |
| **sleuth1714** | .82±.03 | .82±.04 | .81±.14 | **.83±.04** | **.83±.06** | .82±.04 | .76±.03 | .82±.06 | .80±.03 | **.83±.05** |
| **sleuth1605** | .66±.09 | .70±.07 | .64±.08 | .70±.07 | .63±.09 | .66±.05 | .65±.09 | .65±.09 | **.72±.07** | **.72±.05** |
| **sleuth1201** | **.94±.05** | **.94±.03** | .92±.05 | .93±.03 | .91±.05 | .90±.04 | .89±.09 | .88±.06 | .91±.08 | **.94±.05** |
| **rabe266** | .93±.04 | .90±.03 | .91±.04 | .92±.04 | .91±.03 | .92±.03 | .93±.04 | .90±.04 | **.95±.04** | .94±.02 |
| **rabe148** | .95±.04 | .93±.04 | .91±.08 | .95±.04 | .89±.07 | .92±.05 | .91±.06 | .91±.08 | .95±.02 | **.96±.04** |
| **vis_env** | .66±.04 | .68±.05 | .66±.03 | .65±.08 | .62±.04 | .57±.03 | **.69±.06** | .64±.03 | .65±.09 | **.69±.03** |
| **hutsof99** | .74±.07 | .66±.04 | .64±.09 | .73±.07 | .60±.10 | .66±.11 | .66±.14 | .67±.05 | .70±.05 | **.75±.04** |
| **human_dev** | .88±.03 | .85±.04 | .85±.03 | .89±.04 | .85±.03 | .87±.03 | .88±.03 | .86±.03 | .88±.02 | **.89±.04** |
| **c0_100_10** | .77±.04 | .74±.03 | .76±.03 | .77±.03 | .64±.07 | .71±.05 | **.79±.03** | .71±.05 | .78±.01 | .77±.06 |
| **elusage** | .90±.05 | .84±.06 | .84±.06 | .89±.04 | .84±.06 | .87±.05 | .89±.04 | .84±.06 | .89±.04 | **.92±.04** |
| **diggle_table** | .65±.14 | .61±.07 | .57±.08 | .65±.11 | .60±.09 | .58±.07 | .57±.13 | .57±.06 | .60±.13 | **.68±.07** |
| **baskball** | .70±.02 | .68±.04 | .68±.02 | .71±.03 | .71±.03 | .63±.02 | .66±.05 | .69±.04 | .68±.02 | **.72±.06** |
| **michiganacc** | .72±.06 | .67±.06 | .71±.05 | .71±.04 | .67±.06 | .66±.07 | .71±.05 | .69±.04 | .71±.05 | **.73±.05** |
| **election2000** | .92±.04 | .90±.04 | .91±.03 | .92±.02 | .91±.03 | .92±.01 | .90±.07 | .92±.02 | .92±.03 | **.94±.02** |

---

**Algorithm 1** Customized Mirror-Prox algorithm for solving the saddle-point problem (13)

---

Initialize $\epsilon^{(0)}$ and $S^{(0)}$
**for** $t = 0, 1, \ldots, T$ **do**
   Gradient step:
$$\hat{\epsilon}^{(t)} = \text{Proj}_{\mathcal{E}}\left(\epsilon^{(t)} - \alpha_t \nabla_{\epsilon} \phi(\epsilon^{(t)}, S^{(t)})\right)$$
$$\hat{s}_i^{(t)} = \text{Proj}_{\mathcal{S}_i}\left(s_i^{(t)} + \beta_t \nabla_{s_i} \phi(\epsilon^{(t)}, S^{(t)})\right)$$
$$\text{for all } i \in [m]$$

   Extra-gradient step:
$$\epsilon^{(t+1)} = \text{Proj}_{\mathcal{E}}\left(\epsilon^{(t)} - \alpha_t \nabla_{\epsilon} \phi(\hat{\epsilon}^{(t)}, \hat{S}^{(t)})\right)$$
$$s_i^{(t+1)} = \text{Proj}_{\mathcal{S}_i}\left(s_i^{(t)} + \beta_t \nabla_{s_i} \phi(\hat{\epsilon}^{(t)}, \hat{S}^{(t)})\right)$$
$$\text{for all } i \in [m]$$
**end for**
$\hat{\epsilon} = \sum_{t=1}^{T} \alpha_t \hat{\epsilon}^{(t)} / \sum_{t=1}^{T} \alpha_t$
$\hat{S} = \sum_{t=1}^{T} \beta_t \hat{S}^{(t)} / \sum_{t=1}^{T} \beta_t$
Round $\hat{\epsilon}$ to $\{0, 1\}^{p \times n}$
$\hat{W} = -\frac{1}{m\lambda} \sum_{i \in [m]} y_i (\hat{s}_i x_i^T) \odot \hat{\epsilon}$

---

**Algorithm 2** ($\text{Proj}_{\mathcal{E}}$) Projection onto the set $\mathcal{E}_j \triangleq \left\{\epsilon_j \in \mathbb{R}^n : \epsilon_{ji} \in [0, 1], \|\epsilon_j\|_1 \le k\right\}$
Input: $\epsilon_j \in \mathbb{R}^n$ and a small tolerance $tol$.

---

Clip $\epsilon_{j,i}$ to $[0, 1]$ for all $i \in [n]$
Return $\epsilon_j$ if $\mathbf{1}^T \epsilon_j \le k$
   Binary search to find $tol$-solution
Set $low = \left(\mathbf{1}^T \epsilon_j - k\right)/n$
Set $high = \max_{i \in [n]} \epsilon_{j,i} - k/n$
**while** $low \le high$ **do**
   Set $\lambda = (low + high)/2$
   Compute $\hat{\epsilon}_j : \forall i \in [n], \hat{\epsilon}_{j,i} = \epsilon_{j,i} - \lambda$
   Clip $\hat{\epsilon}_j$ to $[0, 1]^n$
   **if** $|\mathbf{1}^T \hat{\epsilon}_j - k| < tol$ **then**
      return $\hat{\epsilon}_j$
   **else if** $\mathbf{1}^T \hat{\epsilon}_j > k$ **then**
      Set $low = (low + high)/2$
   **else**
      Set $high = (low + high)/2$
   **end if**
**end while**

---

In order to use Algorithm 1, we need to find the partial gradients of $\phi(\epsilon, S)$, which are given as

$$\nabla_{\epsilon} \phi(\epsilon, S) = -\frac{1}{m\lambda} \left(\sum_{i \in [m]} y_i \left(s_i x_i^T\right)\right) \odot \left(\sum_{i \in [m]} y_i \left(s_i x_i^T\right)\right),$$

$$\nabla_{s_i} \phi(\epsilon, S) = -\frac{1}{m\lambda} y_i \left(\sum_{i \in [m]} y_i \left(s_i x_i^T\right) \odot \epsilon\right) x_i, \quad i \in [m].$$

There are two projection operators in Algorithm 1. The first one projects some $\epsilon \in \mathbb{R}^{p \times n}$ onto

$$\mathcal{E} \triangleq \left\{\epsilon \in \mathbb{R}^{p \times n} : \epsilon \in [0, 1]^{p \times n}, \|\epsilon\|_{1,\infty} \le k\right\}.$$

This can be done efficiently by Algorithm 2. Essentially, we perform $p$ independent projections, each for one row of $\epsilon$ using a bi-section type of algorithm [39, 40, 41]. We have the following result.

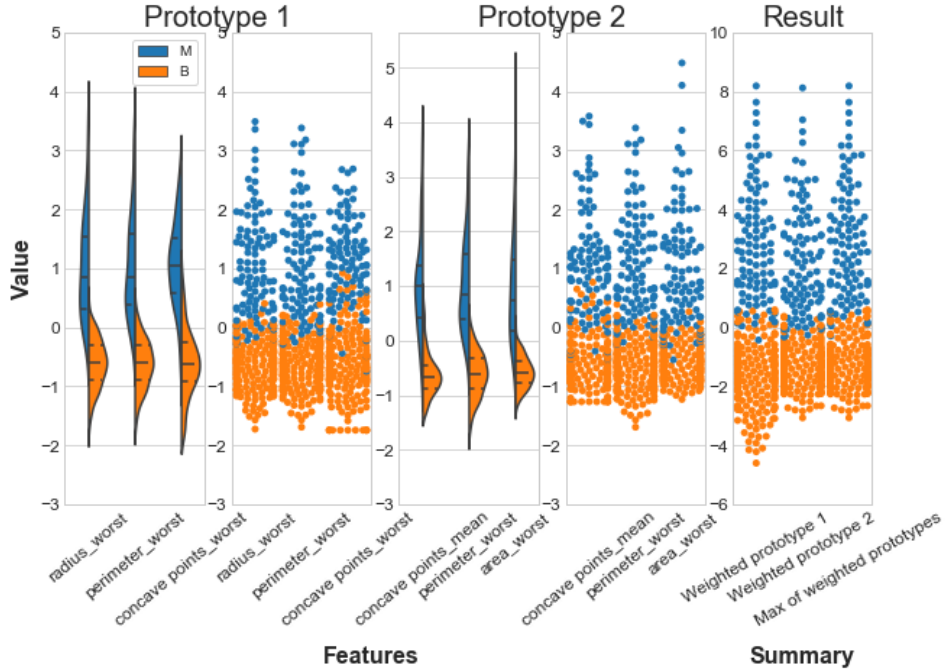

Figure 2: SMaLL applied to the Breast Cancer dataset with $k = 3$ and $p = 2$. The blue and orange dots represent the test instances from the two classes. The plots show the kernel density estimates and the actual values of the non-zero features in each prototype, as well at the final predictor result.

**Proposition 4.** *Algorithm 2 computes, up to a specified tolerance $tol$, the projection of any $\epsilon \in \mathbb{R}^{p \times n}$ onto $\mathcal{E}$ in $\mathcal{O}\left(\log_2(1/tol)\right)$ time, where $tol$ is the input precision for bisection.*

There are two cases for the projection of $s_i \in \mathbb{R}^p$ onto the set $S_i$. For $i \in I_+$, we only need to project $s_{i,j(i)}$ onto the interval $[-1, 0]$ and set $s_{i,j} = 0$ for all $j \neq j(i)$. For $i \in I_-$, the projection algorithm is similar to Algorithm 2, and we omit the details here. The step sizes $\alpha_t$ and $\beta_t$ can be set according to the guidelines described in [37, 38], based on the smoothness properties of the function $\phi(\epsilon, S)$. In practice, we follow the adaptive tuning procedure developed in [42].

# 4 Experiments

We demonstrate the merits of SMaLL via an extensive set of experiments. We start with an intuition into how the class of sparse multiprototype linear predictors differs from standard model classes.

Figure 1 is a visualization of the decision surface of different types of classifiers on the 2-dimensional *chscase funds* toy dataset, obtained from OpenML. The two classes are shown in red and blue, with training data in solid shade and test data in translucent shade. The color of each band indicates the gradation in the confidence of prediction - each classifier is more confident in the darker regions and less confident in the lighter regions. The 2-prototype linear predictor attains the best test accuracy on this toy problem (0.73). Note that some of the examples are highlighted by a black rectangle - the linear classifiers (logistic regression and linear SVM) could not distinguish between these examples, whereas the 2-prototype linear predictor was able to segregate and assign them to different bands.

## 4.1 Low-dimensional Datasets Without Sparsity

We now compare the accuracy of SMaLL with $k = n$ (no sparsity) to the accuracy of other standard classification algorithms, on several low-dimensional ($n \leq 20$) binary classification datasets from the OpenML repository. We experimented with OpenML data for two main reasons: (a) it contains many preprocessed binary datasets, and (b) the datasets come from diverse domains. The methods that we compare against are linear SVM (LSVM), SVM with non-linear kernels such as radial

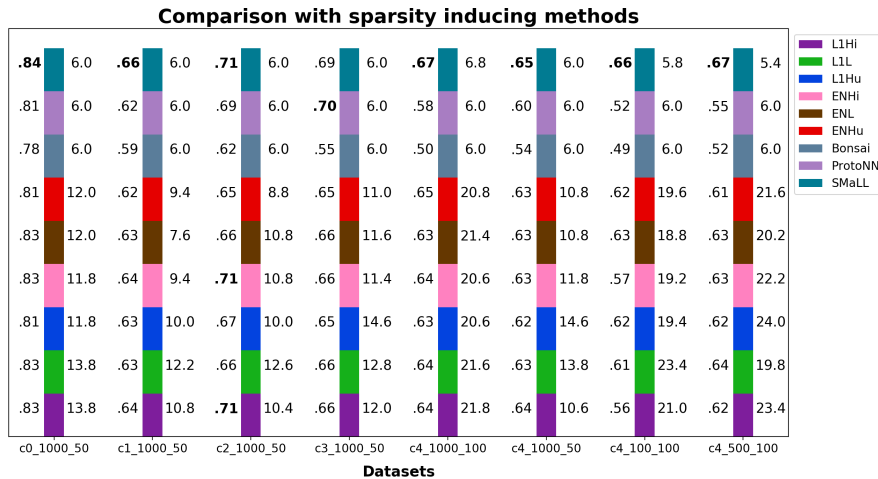

Figure 3: **Comparison on high dimensional** ($n >= 50$) **OpenML data from the *Fri* series.** Each stacked bar shows average test accuracy on left, and the total number of selected features on right.

basis function, polynomial, and sigmoid (RSVM), Logistic Regression (LR), Decision Trees (DT), Random Forest (RF), $k$-Nearest Neighbor (kNN), Gaussian Process (GP), Gradient Boosting (GB), and AdaBoost (AB). All the datasets were normalized to make each feature have zero mean and unit variance. Since the datasets do not specify separate train, validation, and test sets, we measure test accuracy by averaging over five random train-test splits. Since we are interested in extreme sparsity, we pre-clustered the positive examples into $p = 2$ clusters, and initialized the prototypes with the cluster centers. We determined hyperparameters by 5-fold cross-validation. The coefficient of the error term $C$ in LSVM and $\ell_2$-regularized LR was selected from $\{0.1, 1, 10, 100\}$. In the case of RSVM, we also added 0.01 to the search set for $C$, and chose the best kernel between a radial basis function (RBF), polynomials of degree 2 and 3, and sigmoid. For the ensemble methods (RF, AB, GB), the number of base predictors was selected from the set $\{10, 20, 50\}$. The maximum number of features for RF estimators was optimized over the square root and the log selection criteria. We also found best validation parameters for DT (gini or entropy for attribute selection), kNN (1, 3, 5 or 7 neighbors), and GP (RBF kernel scaled with scaled by a coefficient in the set $\{0.1, 1.0, 5\}$ and dot product kernel with inhomogeneity parameter $\sigma$ set to 1). Finally, for our method SMaLL, we fixed $\lambda = 0.1$ and $\alpha_t = 0.01$, and searched over $\beta_t = \beta \in \{0.01, 0.001\}$.

Table 1 shows the test accuracy for the different algorithms on different datasets. As seen from the table, SMaLL with $k = n$ generally performed extremely well on most of these datasets. This substantiates the practicality of SMaLL in the low dimensional regime.

## 4.2   Higher-dimensional Datasets with Sparsity

We now describe results with higher dimensional data, where feature selection becomes especially critical. To substantiate our claim that SMaLL produces an interpretable model, we ran SMaLL on the Breast Cancer dataset with $k = 3$ and $p = 2$ (two prototypes, three non-zero elements in each). Figure 2 shows the kernel density estimates and the actual values of the selected features in each prototype, and the summary of our predictor. Note that the feature *perimeter_worst* appears in both prototypes. As the rightmost plot shows, the predictor output provides a good separation of the test data, and SMaLL registered a test accuracy of over 94%. It is straightforward to understand how the resulting classifier reaches its decisions: which features it relies on and how those features interact.

Next, we compare SMaLL with 8 other methods. Six of these methods induce sparsity by minimizing an $\ell_1$-regularized loss function. These methods minimize one of the three empirical loss functions (hinge loss, log loss, and the binary-classification Huber loss), regularized by either an $\ell_1$ or an elastic net penalty (i.e. $\ell_1$ *and* $\ell_2$). We refer to these as L1Hi ($\ell_1$, hinge), L1L ($\ell_1$, log), L1Hu ($\ell_1$, Huber), EnHi (elastic net, hinge), ENL (elastic net, log) and ENHu (elastic net, huber). We also compare

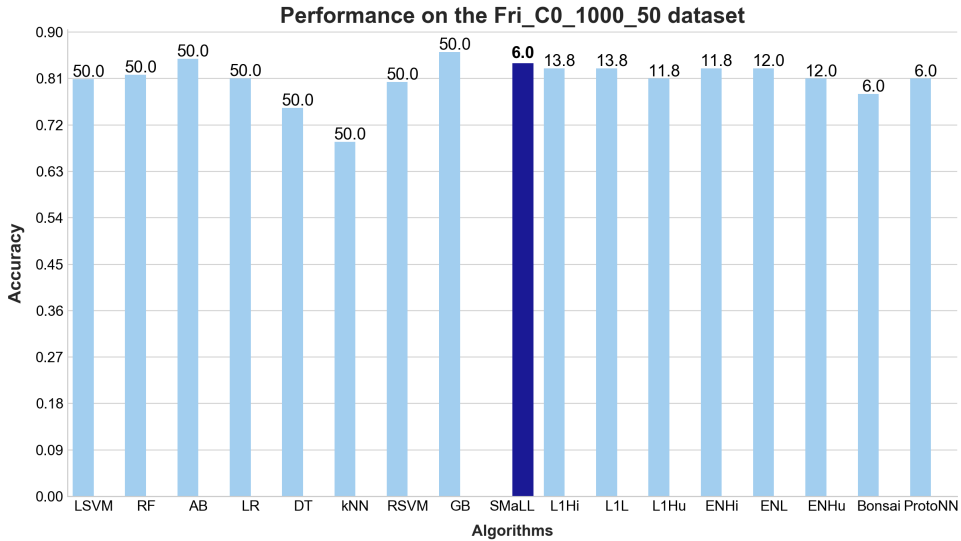

Figure 4: **The big picture.** The plot depicts the performance of SMaLL compared to both the standard classification algorithms and the sparse baselines on the fri_c0_1000_50 dataset. The number atop each bar is the average number of features selected by that algorithm across 5 runs.

with two state-of-the-art methods for the scarce-resource setting. ProtoNN [1] is a modern take on nearest neighbor classifiers, while Bonsai [2] is a sophisticated enhancement of a small decision tree. Note that while we can explicitly control the amount of sparsity in SMaLL, ProtoNN, and Bonsai, the methods that use $\ell_1$ or elastic net regularization do not have this flexibility. Therefore, in order to get the different baselines on the same footing, we devised the following empirical methodology. We specified $p * k = 6$ features as the desired sparsity, and modulated each linear baseline to yield nearly these many features. We trained each of the linear baselines by setting a high value of the $\ell_1$ coefficient and selected the features with the largest absolute values. Then, we retrained the classifier using only the selected features, using the same loss (hinge, loss, or log) and an $\ell_2$ regularization. Our procedure ensured that each baseline benefited, in effect, from an *elastic net*-like regularization while having the most important features at its disposal. For the SMaLL classifier, we fixed $k = 3$ and $p = 2$. In practice, this setting will be application specific (e.g., it would likely depend on the budget). As before, since the original dataset did not specify a train-test split, our results were averaged over five random splits. The parameters for each method were tuned using 5-fold cross-validation. We fixed $\lambda = 0.1$ and performed a joint search over $\alpha_t \in \{0.1, 1e-2, 1e-3\}$ and $\beta_t \in \{1e-3, 1e-4\}$. For all the baselines, we optimized the cross validation error over the $\ell_1$ regularization coefficients in the set $\{1e-1, 1e-2, 1e-3, 1e-4\}$. Moreover, in case of elastic net, the ratio of the $\ell_1$ coefficient to the $\ell_2$ coefficient was set to 1. The depth of the estimators in Bonsai was selected from $\{2, 3, 4\}$. Finally, the dimensionality of projection in ProtoNN was searched over $\{5, 10, 15, 20\}$.

Figure 3 provides strong empirical evidence that SMaLL compares favorably to the baselines on several high dimensional OpenML datasets belonging to the *Fri* series. Specifically, the first number in each dataset name indicates the number of examples, and the second the dimensionality of the dataset. Note that in case of SMaLL, some features might be selected in more than one prototype. Therefore, to be fair to the other methods, we included the multiplicity while computing the total feature count. We observe that, on all but one of these datasets, SMaLL outperformed the ProtoNN and Bonsai models at the same level of sparsity, and the gap between SMaLL and these methods generally turned out to be huge. Moreover, compared to the linear baselines, SMaLL achieved consistently better performance at much sparser levels. This shows the promise of SMaLL toward achieving succinct yet accurate predictors in the high dimensional regime. The merits of SMaLL are further reinforced in Fig. 4 that shows the accuracy-sparsity trade-offs. We observe that just with 6 features, SMaLL provides better test accuracy compared to all the baselines but GB and AB. This shows the potential of SMaLL as a practical algorithm for resource deficient environments.

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
