[Supplementary Material]

# A Supplementary Material

We now provide proofs of the theoretical results stated in the main text.

**Proof of Proposition 1**

*Proof.* Start with a $p$-term $k$-DNF defined over a set of $n$ Boolean variables. Encode the $j$'th term in the DNF formula by a vector $w_j \in \{-1, 0, 1\}^n$, where

$$w_{j,l} = \begin{cases} 1 & l\text{'th variable appears as positive} \\ -1 & l\text{'th variable appears as negative} \\ 0 & l\text{'th variable doesn't appear} \end{cases} . \tag{14}$$

Notice that the resulting vector is $k$-sparse. Next, let $x \in \{-1, 1\}^n$ encode the Boolean assignment of the input variables, where $x_l = 1$ encodes that the $l$'th variable is true and $x_l = -1$ encodes that it is false. Note that the $j$'th term of the DNF is satisfied if and only if $w_j \cdot x \geq k$. Moreover, note that the entire DNF is satisfied if and only if

$$\max_{j \in [p]} w_j \cdot x \geq k , \tag{15}$$

where we use $[p]$ as shorthand for the set $\{1, \ldots, p\}$. We relax this definition by allowing the input $x$ to be an arbitrary vector in $\mathbb{R}^n$ and allowing each $w_j$ to be any $k$-sparse vector in $\mathbb{R}^n$. By construction, the class of models of this form is at least as powerful as the original class of $p$-term $k$-DNF Boolean formulae. Therefore, learning this class of models is a form of improper learning of $k$-DNFs. $\square$

Note that once we allow $x$ and $w_j$ to take arbitrary real values, the threshold $k$ in (15) becomes somewhat arbitrary, so we replace it with zero in our decision rule.

**Proof of Proposition 2**

*Proof.* By definition, the Fenchel conjugate

$$u^\star(s) = \sup_{t \in \mathbb{R}^p} \left( \sum_{k=1}^p s_k t_k - \log \left( 1 + \sum_{k=1}^p \exp(-t_k) \right) \right).$$

Equating the partial derivative with respect to each $t_k$ to 0, we get

$$s_k = -\frac{\exp(-t_k^*)}{1 + \sum_{c=1}^p \exp(-t_c^*)} , \tag{16}$$

or equivalently,

$$t_k^* = -\log \left( -s_k \left( 1 + \sum_{c=1}^p \exp(-t_c^*) \right) \right) .$$

We note from (16) that

$$\frac{1}{1 + \sum_{c=1}^p \exp(-t_c^*)} = 1 + s^\top \mathbf{1} .$$

Using the convention $0 \log 0 = 0$, the form of the conjugate function in (9) can be obtained by plugging $t^* = (t_1^*, \ldots, t_r^*)$ into $u^\star(s)$ and performing some simple algebraic manipulations.

Proposition 2 follows directly from the form of $u^\star$, especially the constraint set $\mathcal{S}_i$ for $i \in I_-$. For $i \in I_+$, we notice that the conjugate of $\ell(z) = \log(1 + \exp(-z))$ is

$$\ell^*(\beta) = (-\beta) \log(-\beta) + (1 + \beta) \log(1 + \beta), \quad \beta \in [-1, 0].$$

Then we can let the $j(i)$th entry of $s_i \in \mathbb{R}^p$ be $\beta \in [-1, 0]$ and all other entries be zero. Then we can express $\ell^*$ through $u^\star$ as shown in the proposition. $\square$

**Proof of Proposition 3**

*Proof.* Recall that

$$\Phi(W, \epsilon, S) = \frac{1}{m} \sum_{i \in [m]} \left( y_i s_i^T (W \odot \epsilon) x_i - u^\star(s_i) \right) + \frac{\lambda}{2} ||W||_F^2.$$

Then

$$\nabla_W \Phi(W, \epsilon, S) = \frac{1}{m} \sum_{i \in [m]} y_i \left( s_i x_i^T \right) \odot \epsilon + \lambda W .$$

The proof is complete by setting $\nabla_W \Phi(W, \epsilon, S) = 0$, and solving for $W$. $\qquad \square$

**Proof of Proposition 4**

*Proof.* In order to project $a \in \mathbb{R}^n$ onto

$$\mathcal{E}_j \triangleq \{\epsilon_j \in \mathbb{R}^n : \epsilon_{ji} \in [0, 1], \ ||\epsilon_j||_1 \leq k\},$$

we need to solve the following problem:

$$
\begin{aligned}
\min_{x \in \mathbb{R}^n} \quad & \frac{1}{2} ||x - a||^2 \\
\text{s.t.} \quad & \sum_{i=1}^d x_i \leq k \\
\forall i \in [n]: \quad & 0 \leq x_i \leq 1 .
\end{aligned}
$$

Our approach is to form a Lagrangian and then invoke the KKT conditions. Introducing Lagrangian parameters $\lambda \in \mathbb{R}_+$ and $u, v \in \mathbb{R}_+^d$, we get the Lagrangian $L(x, \lambda, u, v)$

$$
\begin{aligned}
= \quad & \frac{1}{2} ||x - a||^2 + \lambda \left( \sum_{i=1}^n x_i - k \right) - \sum_{i=1}^n u_i x_i \\
& + \sum_{i=1}^n v_i (x_i - 1) \\
= \quad & \frac{1}{2} ||x - a||^2 + \sum_{i=1}^n x_i (\lambda - u_i + v_i) \\
& - \lambda k - \sum_{i=1}^n v_i .
\end{aligned}
$$

Therefore,

$$\nabla_{x^*} L = 0 \implies x^* = a - (\lambda \mathbf{1} - u + v) . \qquad (17)$$

We note that $g(\lambda, u, v) \triangleq L(x^*, \lambda, u, v)$

$$= -\frac{1}{2} ||\lambda \mathbf{1} - u + v||^2 + a^\top (\lambda \mathbf{1} - u + v) - \lambda K - \mathbf{1}^\top v.$$

Using the notation $b \succeq t$ to mean that each coordinate of vector $b$ is at least $t$, our dual is

$$\max_{\lambda \geq 0, u \succeq 0, v \succeq 0} g(\lambda, u, v) . \qquad (18)$$

We now list all the KKT conditions:

$$
\begin{aligned}
\forall i \in [n]: \quad & x_i > 0 \quad \implies \quad u_i = 0 \\
\forall i \in [n]: \quad & x_i < 1 \quad \implies \quad v_i = 0 \\
\forall i \in [n]: \quad & u_i > 0 \quad \implies \quad x_i = 0 \\
\forall i \in [n]: \quad & v_i > 0 \quad \implies \quad x_i = 1 \\
\forall i \in [n]: \quad & u_i v_i \quad = \quad 0 \\
& \sum_{i=1}^{n} x_i < k \quad \implies \quad \lambda = 0 \\
& \lambda > 0 \quad \implies \quad \sum_{i=1}^{n} x_i = k
\end{aligned}
$$

We consider the two cases, (a) $\sum_{i=1}^{n} x_i^* < k$, and (b) $\sum_{i=1}^{n} x_i^* = k$ separately.

First consider $\sum_{i=1}^{n} x_i^* < k$. Then, by KKT conditions, we have the corresponding $\lambda = 0$. Consider all the sub-cases. Using (17), we get

1. $x_i^* = 0 \implies a_i = \lambda - u_i + v_i = -u_i \leq 0$ (since $x_i^* < 1$, therefore, by KKT conditions, $v_i = 0$).

2. $x_i^* = 1 \implies a_i = 1 + \lambda - u_i + v_i = 1 + v_i \geq 1$ (since $x_i^* > 0$, therefore, $u_i = 0$ by KKT conditions).

3. $0 < x_i^* < 1 \implies a_i = x_i^* + \lambda - u_i + v_i = x_i^*$.

Now consider $\sum_{i=1}^{n} x_i^* = k$. Then, we have $\lambda \geq 0$. Again, we look at the various sub-cases.

1. $x_i^* = 0 \implies a_i = \lambda - u_i + v_i = \lambda - u_i \implies u_i = -(a_i - \lambda)$. Here, $u_i$ denotes the amount of clipping done when $a_i$ is negative.

2. $x_i^* = 1 \implies a_i = 1 + \lambda - u_i + v_i = 1 + \lambda + v_i \implies v_i = -(1 + \lambda - a_i)$. Here, $v_i$ denotes the amount of clipping done when $a_i > 1$. Also, note that $a_i \geq 1$ in this case.

3. $0 < x_i^* < 1 \implies a_i = x_i^* + \lambda - u_i + v_i = x_i^* + \lambda \implies x_i^* = a_i - \lambda$. In order to determine the value of $\lambda$, we note that since $\sum_{i=1}^{n} x_i^* = k$, therefore,

$$
\sum_{i=1}^{n} (a_i - \lambda) = k \implies \sum_{i=1}^{n} a_i - n\lambda = k
$$

$$
\implies \lambda = \frac{1}{n} \sum_{i=1}^{n} a_i - \frac{k}{n} \leq \max_i a_i - \frac{k}{n} .
$$

Algorithm 2 implements all the cases and thus accomplishes the desired projection. The algorithm is a bisection method, and thus converges linearly to a solution within the specified tolerance $tol$. $\square$