[Reviews · NeurIPS 2018]

Reviewer 1



The paper proposes a new algorithm for the training of multi-prototype binary classifier by solving its convex-concave saddle-point relaxation. Both strong theoretical result and comprehensive empirical study are provided. More detailed comments: (i) model size (timing result): the title of the paper is "SMaLL", but the model sizes or timing results are not provided as related works that also experiment on the multi-prototype model (Gupta et al, 2017) did. Only number of selected features is reported, which is not enough when comparing with other types of models like sparse logistic regression (since it only has one vector while the proposed model has multiple vectors). (ii) algorithmic improvement vs. model improvement: it would be good to have comparison to other optimization algorithms for the multi-prototype classifier, as for now from the results we only see the joint effect of algorithm and model. Is the improved accuracy due to a better algorithm? or a better model? Is there existing work that learns the same (or similar) model with a different algorithm? (iii) The data sets are from openML and do not contain common datasets with some previous works such as (Gupta et al, 2017). This makes it difficult to check consistency of the accuracy results with other papers. Could the authors provide links to some papers that also use the OpenML datasets during the author feedback? And also why those datasets are chosen? Also, for high-dimensional experiments, it seems only one data set of different settings is used. Why is it the case?

Reviewer 2



Summary of paper: The paper addresses the problem of learning in resource constrained setting using multiple linear predictors (max of linear predictors like DNF). The problem of learning k-DNFs (a subset of multiple linear predictord) is known to be hard however to solve this non-convex problem, they use various relaxations to formulate this as a minmax saddle point objective which they solve using standard mirror-prox algorithm. The paper then goes on to show that the approach is fruitful in practice through experiments. Comments for authors: In the introduction (line 28-31), you seem to sugest that you can solve the problem of k-DNFs however you do not actually give any guarantees for the final learnt hypothesis that your algorithm outputs, please make that clear. Requiring a different weight vector for each positive example, this seems very costly and potentially avoidable. It would be good to discuss how the clustering affects this, probably show a plot for different cluster sizes vs performance. Overall comments: Using Multiple linear predictors as the base hypothesis class is a good direction as it is a richer class of functions compared to standard sparse linear regressors (LASSO). Viewing the problem as a minmax saddle point problem is an interesting take and I find the use of the boolean constraint matrix novel. As is, the algorithm is not efficient (dimension of weight vector can be of the order of the number of samples) and they do not give any theoretical guarantees for the hypothesis returned by their algorithm. However, the experimental results do seem to suggest that in practice this approach gives a compact sparse representation with low computational overhead. Overall, the paper is well written, provides a useful framework for solving a common problem and from a practical standpoint gives gains over most standard methods (not always large) while giving a handle on the exact sparsity required in the final hypothesis.

Reviewer 3



This paper considers a multi-prototype model for binary classification problem, aiming to find a compact and interpretable model. The problem can then be relaxed and transformed into a smooth convex-concave saddle-point problem, which can be solved by Mirror Prox (a variant of Mirror Descent) algorithm generalized for saddle points. For experiments, this approach outperforms state-of-the-art baselines in various datasets. Strengths: The formulation of convex-concave saddle-point problem is interesting. Extensive experiments are conducted. Weaknesses: 1. It is confusing to me what the exact goal of this paper is. Are we claiming the multi-prototype model is superior to other binary classification models (such as linear SVM, kNN, etc.) in terms of interpretability? Why do we have two sets of baselines for higher-dimensional and lower-dimensional data? 2. In Figure 3, for the baselines on the left hand side, what if we sparsify the trained models to reduce the number of selected features and compare accuracy to the proposed model? 3. Since the parameter for sparsity constraint has to be manually picked, can the authors provide any experimental results on the sensitivity of this parameter? Similar issue arises when picking the number of prototypes. Update after Author's Feedback: All my concerns are addressed by the authors's additional results. I'm changing my score based on that.